# Guidelines for Management of Urgent Symptoms in Patients with Cholangiocarcinoma and Biliary Stents or Catheters Using the Modified RAND/UCLA Delphi Process

**DOI:** 10.3390/cancers12092375

**Published:** 2020-08-21

**Authors:** Renuka V. Iyer, Susan G. Acquisto, John A. Bridgewater, Michael A. Choti, Theodore S. Hong, Bela Kis, Peter A. Mead, Neehar D. Parikh, Lewis R. Roberts, Rebecca Roberts, Riad Salem, Jason K. Sicklick, Richard S. Siegel, Jonathan R. Whisenant, Dasha Cherepanov, Michael S. Broder, Juan W. Valle

**Affiliations:** 1Roswell Park Cancer Institute, Buffalo, NY 14263, USA; renuka.iyer@roswellpark.org; 2Cholangiocarcinoma Foundation, Herriman, UT 84096, USA; sacquisto@rrcsinc.com; 3University College London Cancer Institute, London WC1E 6DD, UK; j.bridgewater@ucl.ac.uk; 4Banner MD Anderson Cancer Center, Gilbert, AZ 85234, USA; Michael.choti@bannerhealth.com; 5NSABP/NRG Oncology and Massachusetts General Hospital, Boston, MA 02114, USA; TSHong1@mgh.harvard.edu; 6Department of Interventional Radiology, Moffitt Cancer Center, Tampa, FL 33612, USA; bela.kis@moffitt.org; 7Memorial Sloan Kettering Cancer Center, New York, NY 10065, USA; meadp@mskcc.org; 8Division of Gastroenterology, Department of Internal Medicine, University of Michigan Health System, Ann Arbor, MI 48109, USA; ndparikh@med.umich.edu; 9Mayo Clinic, Rochester, MN 55902, USA; roberts.lewis@mayo.edu; 10Cook County Health and Hospital System, Chicago, IL 60612, USA; Rebeccarroberts55@gmail.com; 11Department of Radiology, Northwestern University, Chicago, IL 60208, USA; r-salem@northwestern.edu; 12Department of Surgery, University of California San Diego, La Jolla, CA 92093, USA; jsicklick@ucsd.edu; 13NW Medical Consult, Glenview, IL 60026, USA; richard.siegel@usoncology.com; 14Intermountain Healthcare, Murray, UT 84107, USA; jonathan.whisenant@hci.utah.edu; 15Partnership for Health Analytic Research, LLC, Beverly Hills, CA 90212, USA; dgcherepanov@gmail.com (D.C.); mbroder@pharllc.com (M.S.B.); 16Division of Cancer Sciences and Department of Medical Oncology, University of Manchester, The Christie NHS Foundation Trust, Manchester M13 9PL, UK

**Keywords:** cholangiocarcinoma, biliary cancer, Delphi, biliary emergencies, PTC, biliary stent

## Abstract

*Background*: Patients with cholangiocarcinoma often have indwelling biliary stents or catheters which are prone to obstructions and/or infections; studies show that 20–40% present with fever and/or jaundice requiring urgent treatment in the outpatient setting for which there are no uniform guidelines. The goal was to develop an expert panel consensus on this topic using the modified RAND/UCLA Delphi process to rate treatment appropriateness. *Methods*: Thirteen expert physicians from relevant specialties, geography, and practice settings were recruited for the panel. Patient scenarios were developed and panelists rated the therapies before and after a face-to-face discussion. The appropriateness of various therapies was rated on a scale from 1–9 and classified as appropriate, inappropriate, or uncertain. Scenarios with greater than 2 (>2) ratings of 1–3 (inappropriate) and greater than 2 (>2) ratings of 7–9 (appropriate) were considered to have disagreement and were not assigned an appropriateness rating. *Results*: Panelists were from all US regions and the UK (8%) and had practiced for a mean 16.5 years (4–33 years). Panelists rated 480 scenarios before the meeting and re-rated 288 of the clinical scenarios after the meeting. The panelists agreed that ongoing treatment with chemotherapy did not influence decision-making and, therefore, 192 scenarios were excluded from the final list. Disagreement decreased from 37.5% before to 10.4% after the meeting. Consensus on stent/tube manipulation and inpatient antibiotic therapy was obtained and summarized in patients as “appropriate” or “maybe appropriate” based on a patient’s bilirubin level at presentation. *Conclusions*: The Delphi process produced consensus guidelines to fill an unmet need in the urgent management of ascending cholangitis in patients with cholangiocarcinoma.

## 1. Introduction

Cholangiocarcinoma (CC) is a cancer arising from the intra- or extra-hepatic bile ducts, mainly characterized by its late diagnosis and fatal outcome [1]. Painless obstructive jaundice is a common presenting symptom and managing the malignant biliary obstruction to ensure adequate drainage is necessary to effectively treat both cholangiocarcinoma and a subset of pancreatic cancer patients (those with disease in the head of the pancreas). Patients with cholangiocarcinoma are at a higher risk of developing ascending cholangitis once instrumented, stented, or following the placement of a percutaneous drain. In the Advanced Biliary Cancer (ABC)-02 study 45% of patients had biliary stents [2]. Almost all extrahepatic CCA patients present with biliary obstruction and require stent placement. The rate of infection in malignant obstruction is reported to be anywhere from 20–40% [3,4]. The ABC-02 study reported an infection rate of 19% [2]. Some centers have reported that iatrogenic biliary intervention has replaced choledocholithiasis as the most common cause of severe cholangitis [5]. Management options include medical, endoscopic, percutaneous, and surgical therapeutic options as well as antibiotics and hospitalization, depending on a variety of factors [6]. There are guidelines for the management of biliary cancers that acknowledge the need for palliative measures, however, no consistency currently exists in recognition and management of ascending cholangitis [7,8,9,10].

The Cholangiocarcinoma Foundation (CCF) Nursing Advisory Board (NAB) conducted an internal comprehensive review of literature relating to supportive care and symptom management of malignancies affecting the biliary tree. The CCF NAB, in conjunction with clinical experts, determined that there are no accepted recommendations for the treatment of biliary emergencies in CC patients with biliary stents or catheters. To meet this need, the CCF organized a RAND/UCLA modified Delphi panel to develop consensus guidelines for the management of patients diagnosed with CC who have biliary stents or catheters in place and present for emergency management in the acute outpatient setting [11].

To conduct this panel, a series of patient scenarios were developed aimed at reflecting everyday clinical practice. The scenarios describe CC patients with biliary stents or catheters presenting with clinical or laboratory abnormalities to an acute outpatient setting (e.g., office visit, emergency department). Each patient scenario was rated at least once by each expert panelist—once prior to the in-person meeting and once at the meeting. The second-round aggregate ratings were used to develop statements reflecting the expert panel consensus.

## 2. Methods

### 2.1. Delphi Process

A RAND/UCLA modified Delphi expert panel process was conducted [11], which included the following sequential steps:(1)Development of the theoretical framework(2)Identification of panelists(3)Review and summary of literature(4)Development of a rating form(5)Completion of the rating form by panelists before an in-person meeting (first-round ratings)(6)Discussion of the initial rating results between panelists at the meeting(7)Completion of the rating form by each panelist a second time (second-round ratings)(8)Development of consensus statements based on second round results

### 2.2. Theoretical Framework Development

The steering committee members, who also served on the panel, were responsible for the scientific integrity of the entire process (Table 1). The committee members refined the specific goals of the meeting, selected panel members, and oversaw the process. In collaboration with the panel and steering committee members, the Partnership for Health Analytic Research (PHAR) reviewed existing literature related to patients with malignancies that affect the biliary ducts. A theoretical framework that described the relevant clinical and therapeutic domains was developed with the goal of ensuring that the Delphi panel addressed the issues of greatest relevance to clinicians treating emergencies in patients with malignancies that affect the biliary ducts.

### 2.3. Panelist Identification

PHAR worked with the steering committee chair (RI) and CCF to develop a list of panelist knowledge areas and characteristics that must be represented. These characteristics accounted for wide variation in health care systems across the United States and United Kingdom and organized the panel around a list of key attributes. These attributes included practice type and setting, geographic variation, and physician specialty. After developing the attribute list, the steering committee chair and CCF identified individuals practicing with these qualities and invited them to participate in this panel.

### 2.4. Evidence Summarization

The CCF NAB previously conducted an internal comprehensive review of literature relating to supportive care and symptom management of malignancies affecting the biliary ducts. This review was provided to the PHAR team, which they supplemented with an additional targeted review. A summary of published evidence relevant to the topic was developed. The purpose of the literature summary was to provide relevant comprehensive information to inform the completion of the rating form (to be discussed next). In the weeks leading up to the meeting, PHAR distributed the relevant literature to panel members, along with a bibliography and the literature summary document.

### 2.5. Detailed Rating Survey Development

The modified Delphi expert panel process required obtaining a formal survey of panelists opinions before the meeting. These first-round survey results were the central focus of the in-person discussion. After the meeting, the survey was repeated by each panelist.

Rating form development was a complex process that involved the interplay of typical prescribing practices, cognitive styles of prescribers, and mechanics of describing multiple different patient types in an interpretable and concise way. This was an iterative process, requiring multiple rounds of review and revision with input from the steering committee and panelists.

A detailed survey of opinions on key issues of interest to be completed by all panelists after reviewing the evidence summary was developed. The survey provided a quantitative assessment of the panelists’ thoughts on the key topics before meeting in-person. Specifically, a series of clinical patient scenarios and a series of potential management options were developed. The scenarios described CC patients with biliary stents or catheters presenting with clinical or laboratory abnormalities to an outpatient acute care setting.

Each scenario was rated on a scale from 1 to 9 with respect to its appropriateness for the situation and weight of evidence supporting its use. Panelists were instructed to consider appropriate therapy as treatment for which the expected health benefit outweighs the expected harms (e.g., side effects, any inconveniences or recovery burdens associated with use of a particular therapy) by a wide enough margin that it is worth doing, without consideration of cost. A rating of 1 was used to indicate an inappropriate treatment, meaning one for which the expected harms greatly outweighed the expected benefits. A rating of 5 indicated either an equal trade-off between harms and benefits or that the scenario cannot be rated for appropriateness. A rating of 9 indicated an appropriate treatment for which expected benefits greatly outweighed the expected harms.

### 2.6. Survey Administration

PHAR distributed the summary document and rating form to panelists, collected it, and then analyzed the results before the in-person meeting with all the panelists. Panel responses were collected using an interactive rating form. Before completing the survey, the panelists were expected to read the literature review summary of evidence provided to them. During this process, each panelist was contacted to ensure that they understood the rating form assumptions, definitions, and instructions. At the conclusion of the in-person meeting, each panelist submitted their second-round responses.

### 2.7. Statistical Analysis

Summary statistics for the first-round ratings were conducted in Microsoft Excel (Microsoft, Redmond, WA, USA). For both the pre-meeting and second rounds, the median of the panelists ratings and mean absolute deviation from each panelist’s rating from the median was calculated for each patient scenario. The median value was used to measure central tendency because the responses were ordinal and the distance between points on the scale was not fixed (e.g., an 8 and a 9 might be closer together than a 4 and a 5). Average distance from the median was used to measure dispersion.

Whether respondents agreed or disagreed on each treatment scenario was also reported. For example, median rating (7–9) indicated agreement that the treatment was appropriate, median rating (1–3) indicated agreement that the treatment was inappropriate, and median rating (4–6) indicated panel agreement that it is uncertain whether the treatment is appropriate or inappropriate. Statistics summarizing mean appropriateness and frequency of agreement (e.g., not at the individual patient scenario level, but over entire rating form sections) were also reported.

The output from the panel was analyzed the same way as were the pre-meeting ratings. Once again, each indication was categorized as appropriate, inappropriate, or uncertain. The appropriateness of classifications and summary statistics (i.e., medians and absolute deviations) for each scenario were summarized for the entire form.

### 2.8. Delphi Panel Meeting

The face-to-face Delphi panel meeting, guided by impartial moderators, took place on Friday, 3 July 2016, from 7:00 AM to 1:00 PM, at Renaissance Blackstone Chicago (636 South Michigan Avenue, Chicago, IL, USA).

At the meeting, the moderators gave a brief presentation describing the meeting goals, the RAND/UCLA modified Delphi process, and the form assumptions, definitions, and instructions. The rest of the meeting was spent with the panel discussing each patient scenario and reviewing the content of their first-round ratings, which resulted in either three levels of agreement (inappropriate treatment, appropriate treatment, or uncertainty regarding appropriateness) or disagreement.

The goal of this discussion was not to have the panel reach a formal consensus during the meeting, but to allow the panel an exhaustive opportunity to achieve a common ground in terms of understanding the form goals, structure, assumptions, definitions, and instructions. At the conclusion of the in-person meeting, panelists completed the survey a second time. These second-round ratings provided the basis for the panel treatment consensus statements.

### 2.9. Development of Consensus Statements

The recommendations resulting from the second-round survey results represented the group consensus and formed the basis for the final treatment recommendations. Indications for management options for which there was a high level of agreement were transformed into declaratory statements that made recommendations based on the panel’s interpretation of the evidence.

## 3. Results

Given the objective to develop consensus guidelines on the management of CC patients with a biliary stent or catheter in place, and present for emergency management in the outpatient acute setting, some key assumptions were made and agreed upon. The consensus statements that follow apply to patients under the assumption that the patient:(1)is out of the immediate post-operative period from any surgical procedures.(2)has access to necessary care (e.g., insurance coverage, experienced physicians).(3)can be transferred to higher level care if necessary.(4)has not signed a do not resuscitate order and is not terminal.(5)is not awaiting liver transplantation.(6)is given symptomatic treatment (e.g., pain medications, IV fluids), palliative treatment (e.g., palliative surgery), counseling, and emotional support as needed.(7)will have his/her disease-directed treatments modified (e.g., from one chemotherapeutic agent to another, from chemotherapy to radiation/liver directed therapy) by an expert oncologist/other specialist after the acute situation is resolved.(8)has had all tests necessary to make therapeutic recommendations including imaging of the biliary tract (e.g., by ultrasound, CT or MR) according to institutional guidelines.(9)recommendations for antibiotics do not address peri-procedural use, which is clinician and institution dependent.

The panel recognized that significant heterogeneity will remain within each scenario and recommends that physicians use clinical judgment when applying any of these consensus statements to patient care. Like in the Tokyo guidelines, the panelists felt the key drivers of decision making were felt to be the bilirubin, liver function abnormalities, white blood cell count, fever, presence/absence of new or worsening biliary dilatation by imaging, and ECOG performance score. The key interventions with this clinical and laboratory data were inpatient antibiotics, outpatient antibiotics, and stent/PTC manipulation or replacement (Table 2).

During the meeting, the panel agreed that recommendations for 192 scenarios would not differ by whether or not patients were actively being treated with chemotherapy. As a result, 192 scenarios were not re-rated in the second round. The remaining 288 scenarios were re-rated by 13 panelists (one panelist was unavailable for round 2). Panelists rated each treatment scenario and 87 (30.2%) of treatment options presented were rated as inappropriate, for 75 (25.3%) scenarios panelists were uncertain as all options were reasonable and/or there was no data to select one choice over another, and 98 (34%) treatment scenarios were felt by the majority to be appropriate. The definition of inappropriate, uncertain, appropriate, and disagreement were standard for the Delphi process (Table 3). Scoring agreement was high and is summarized in Table 4. Disagreement decreased from 37.5% in the first round to 10.4% (30 scenarios) after the panel meeting. Consensus statements resulting from the second round are listed below and summarized in Table 5.

Consensus Statements:(A)In a patient with elevated bilirubin:(1)Stent manipulation is appropriate.(2)Inpatient antibiotics are appropriate if the patient is febrile.(a)Outpatient antibiotics may be an appropriate alternative in patients with no new or worsening biliary dilatation, unless they have neutropenia.(3)Inpatient antibiotics may be appropriate if the patient is afebrile but has elevated WBC and/or is neutropenic(a)Uncertain if outpatient antibiotics may be an appropriate alternative.(4)Antibiotics are inappropriate if the patient is afebrile and has normal WBC.(B)In a patient with normal bilirubin:(1)Stent manipulation(a)is appropriate if the patient has new or worsening biliary dilatation, fever, and good performance status.(b)is inappropriate if the patient has no new or worsening biliary dilation.(2)Inpatient antibiotics are appropriate if the patient is febrile.(a)Outpatient antibiotics may be an appropriate alternative unless the patient is neutropenic.(3)Antibiotics (either inpatient or outpatient) may be appropriate if the patient is afebrile but has new or worsening biliary dilatation and is neutropenic or has elevated WBC.(4)Antibiotics are inappropriate if the patient is afebrile and has a normal WBC.

The panel recognized that significant heterogeneity will remain within each scenario and recommends that physicians use clinical judgment when applying any of these consensus statements to patient care.

## 4. Discussion

Malignant biliary obstruction and biliary procedures render patients at greater risk for ascending cholangitis that can be fatal if not detected and treated appropriately [12,13]. The Tokyo Guidelines are a validated set of diagnostic criteria used as a standard for the diagnosis of acute cholangitis. The criteria have three components: evidence of systemic inflammation (fever or elevated WBC count), cholestasis (elevated bilirubin or transaminases), and evidence of etiology on imaging. The Tokyo guidelines reviewed the available literature on blood and bacterial cultures taken from patients with various biliary diseases, including post-operative patients, all of which emphasized the importance of early antimicrobial therapy that cover intestinal bacterial flora such as *Escherichia coli*. *Klebsiella*, *Enterococcus faecalis*, and *Enterobacter* which were the most frequently isolated bacterial species [14,15,16,17]. *Streptococcus spp.*, *pseudomonas*, and *Proteus* were less frequently seen and anaerobes were isolated more often in the context of polymicrobial infection [17,18]. The resulting guidelines provide grading and severity, but do not provide consensus recommendations on intervention for management in the setting of cancer. A randomized trial including 54 patients was conducted in the preoperative setting to determine which form of biliary drainage is ideal (percutaneous versus endoscopic). This was a randomized prospective study to report cholangitis rates in an operable patient cohort to be 16 (59%) in the percutaneous transhepatic biliary drainage group and 10 (37%) in the endoscopic biliary drainage group (*p* = 0.1) [19]. In patients requiring drainage with complex hilar structures, an intent should be made for endoscopic drainage with the use of percutaneous drainage only when necessary given not just the higher morbidity but also mortality [20].

The Delphi method used in this study is based on the principle that forecasts or decisions from a structured group of individuals are more accurate than those from unstructured groups. This method has been used since the 1950s where there is a lack of definitive methods for conducting research and lack of statistical support for the conclusions drawn. Including appropriate specialties from a variety of proactive settings and geographic regions, the minimum required data for decision making when a patient presents with urgent symptoms with biliary stents or catheters was the availability of stent/tube manipulation, basic blood tests, and imaging to assess if the patient has new or worsening biliary ductal dilatation. This was, of course, in addition to basic symptoms and the clinical picture. The main drivers of decisions were bilirubin, WBC count, presence of new or worsening biliary dilation, and whether the patient was febrile or not. There was consensus in manipulating the stent/tube where the bilirubin was elevated but in the setting of normal bilirubin, this was felt to be appropriate only if there was new or worsening biliary dilation.

Inpatient antibiotics were felt to be appropriate if the patient is febrile, regardless of bilirubin. There was also consensus that inpatient antibiotics would be inappropriate if the patient is afebrile and has a normal WBC. In patients with elevated bilirubin, inpatient antibiotics may be appropriate if the patient is afebrile but has an elevated WBC or is neutropenic. However, in patients with normal bilirubin, inpatient antibiotics may be appropriate if the patient is afebrile and has an elevated WBC or is neutropenic along with new or worsening biliary dilatation. Of note, the panel did not generate any clear recommendations on outpatient antibiotics.

The scenarios to generate recommendations were initially developed both in the context of a patient on chemotherapy and not on treatment but after the second round of scoring, the group found the decisions were similar as long as the patient was pursing aggressive palliative care for their cancer. Therefore, active treatment did not dictate management decisions of urgent symptoms in patients with cholangiocarcinoma and biliary stents or catheters.

In patients with malignancy, prophylactic antibiotics use is only recommended peri-procedure due to transient bacteremia after tube and stent change. This is the first consensus guideline on use of antibiotics in the setting of urgent symptoms in patients with stents or tubes. In clinic practice, it is challenging to distinguish chemotherapy-related fever and drug-related or stent related liver function test (LFT) abnormalities from cholangitis. The panelists were asked to rank which lab abnormalities, clinical symptoms, and radiographic findings drove decision making and this guide is easy to use given the consensus from experts on using bilirubin, temperature, new or worsening biliary dilation, and WBC count which should be helpful to clinicians. Neutropenia rates on chemotherapy are 25% on gemcitabine with cisplatin and 45% with FOLFIRINOX, the two commonly used chemotherapy regimens in patients with biliary stents or drains and these patients may not have a WBC spike due to myelosuppression. In these patients, changes in LFTs can be the only laboratory changes which indicate acute cholangitis. Abnormal LFTs are not often compared by emergency room doctors to baseline abnormalities, in part because of a lack of ready access to prior laboratory results as well as the levels often assumed to be secondary to malignancy or treatment rather than infection. Continued patient education to use the Cholangiocarcinoma Foundation developed Biliary Emergency Card (available at cholangiocarcinoma.org) may allow communication between the emergency care providers and primary oncology team to have a high index of suspicion for cholangitis and better care for these patients. Evaluation of the clinical benefit of these recommendations was outside the scope of the project; future studies measuring the outcomes of patients managed according to these recommendations are warranted.

In conclusion, we present the first consensus recommendations from all subspecialists who encounter these patients to guide treatment and ensure timely and appropriate intervention. The assumptions made to allow decision making were that the liver functions were able to be compared to baseline, appropriate imaging to identify new biliary obstruction were available, and the center had access for stent tube or other manipulation as required. While many factors are taken into consideration, the experts felt that the white count, bilirubin, and fever were the most weighted in deciding stent manipulation and inpatient versus outpatient antibiotics.

These recommendations merit prospective validation and cost effectiveness analyses to measure impact on outcomes and overall care.

## Figures and Tables

**Table 1 cancers-12-02375-t001:** Panelists.

Name	Affiliation	Discipline	Role
Susan Acquisto, DNP, RN, NEA-BC	Cholangiocarcinoma Foundation, Herriman, UT, USA	Nursing	Steering Committee, panelist
John Bridgewater, MRCP, PhD	University College of London Hospitals, London, UK	Medical Oncology	Steering Committee, panelist
Michael Choti, MD, MBA	Banner MD Anderson Cancer Center, Gilbert, AZ, USA	Surgical Oncology	Steering Committee, panelist
Theodore Hong, MD	Dana-Farber/Harvard University, Boston, MA, USA	Radiation Oncology	Steering Committee, panelist
Renuka Iyer, MD	Roswell Park Cancer Institute, Buffalo, NY, USA	Medical Oncology	Steering Committee Chair, panelist
Bela Kis, MD, PhD	Moffitt Cancer Center, Tampa, FL, USA	Interventional Radiology	Panelist
Peter Mead, MD	Memorial Sloan Kettering Cancer Center, New York, NY, USA	Infectious Diseases	Panelist
Neehar Parikh, MD	University of Michigan Health System, Ann Arbor, MI, USA	GI-Veterans Affairs	Panelist
Lewis Roberts, MB ChB, PhD	Mayo Clinic, Rochester, MN, USA	Hepatology	Steering Committee, panelist
Rebecca Roberts, MD	Cook County Health and Hospital System, Chicago, IL, USA	Emergency Medicine	Panelist
Riad Salem, MD, MBA	Northwestern University, Chicago, IL, USA	Vascular and Interventional Radiology	Panelist
Richard Siegel, MD	Advocate Lutheran General Hospital, Arlington Heights, IL, USA	Community Oncologist	Panelist
Jason Sicklick, MD	University of California, San Diego, CA, USA	Surgical Oncology	Panelist
Juan Valle, MB ChB, MSc, FRCP	University of Manchester/The Christie NHS Foundation Trust, Manchester, United Kingdom	Medical Oncology	Steering Committee Chair, panelist
Jonathan Whisenant, MD	Intermountain Medical Center, Murray, UT, USA	Hematology	Steering Committee, panelist

Moderators: Michael S. Broder, MD, MSHS (president) and Dasha Cherepanov, PhD (Director-Outcomes Research) Partnership for Health Analytic Research, LLC. Cholangiocarcinoma Foundation Observers: Stacie Lindsey (President) and Donna Mayer (Executive Director).

**Table 2 cancers-12-02375-t002:** Results according to patient scenario. In every cell below, indicate the appropriateness of therapy on a scale 1 to 9.

Rate the Appropriateness of Each Therapy:				No New or Worsening Biliary Dilatation by Imaging						New or Worsening Biliary Dilatation by Imaging					
				ECOG 0–2			ECOG 3			ECOG 0–2			ECOG 3		
				Inpt Abx	Outpt Abx	Stent Manipu-Lation	Inpt Abx	Outpt Abx	Stent Manipu-Lation	Inpt Abx	Outpt Abx	Stent Manipu-Lation	Inpt Abx	Outpt Abx	Stent Manipu-Lation
**Normal bilirubin**	**Normal ALT/AST**	**Febrile**	**Neutropenia**	A1 9 (0.3)	A2 5 (1.3)	A3 1 (0.2)	A4 9 (0.0)	A5 4 (1.5)	A6 1 (0.0)	A7 9 (0.1)	A8 2 (0.8)	A9 7 (2.2)	A10 9 (0.2)	A11 2 (1.5)	A12 7 (2.2)
			**Normal WBC**	B1 5 (1.2)	B2 5 (0.7)	B3 1 (0.2)	B4 7 (1.1)	B5 5 (0.5)	B6 1 (0.0)	B7 7 (1.2)	B8 5 (1.2)	B9 8 (2.1)	B10 8 (1.0)	B11 5 (0.7)	B12 5 (1.5)
			**Elevated WBC**	C1 8 (1.0)	C2 6 (0.9)	C3 1 (0.5)	C4 8 (0.8)	C5 5 (0.7)	C6 1 (0.0)	C7 9 (0.5)	C8 5 (1.3)	C9 9 (1.8)	C10 9 (0.5)	C11 5 (1.2)	C12 6 (1.9)
		**Afebrile**	**Neutropenia**	D1 2 (0.9)	D2 3 (1.7)	D3 1 (0.0)	D4 2 (1.5)	D5 3 (1.5)	D6 1 (0.1)	D7 5 (1.4)	D8 5 (1.7)	D9 5 (2.5)	D10 5 (2.0)	D11 5 (0.9)	D12 3 (2.2)
			**Normal WBC**	E1 1 (0.0)	E2 1 (0.0)	E3 1 (0.0)	E4 1 (0.0)	E5 1 (0.0)	E6 1 (0.0)	E7 1 (0.0)	E8 1 (0.0)	E9 5 (2.8)	E10 1 (1.8)	E11 1 (0.8)	E12 3 (2.1)
			**Elevated WBC**	F1 2 (0.9)	F2 5 (1.2)	F3 1 (0.0)	F4 3 (1.3)	F5 5 (0.7)	F6 1 (0.0)	F7 5 (1.9)	F8 5 (1.5)	F9 5 (2.5)	F10 5 (2.2)	F11 4 (1.3)	F12 5 (2.0)
	**Elevated ALT/AST**	**Febrile**	**Neutropenia**	G1 9 (0.3)	G2 5 (1.9)	G3 1 (0.2)	G4 9 (0.0)	G5 2 (1.5)	G6 1 (0.0)	G7 9 (0.1)	G8 2 (0.7)	G9 8 (1.9)	G10 9 (0.2)	G11 2 (1.7)	G12 7 (1.8)
			**Normal WBC**	H1 6 (1.5)	H2 5 (0.7)	H3 1 (0.3)	H4 7 (1.2)	H5 5 (0.5)	H6 1 (0.0)	H7 7 (1.1)	H8 5 (1.1)	H9 7 (2.0)	H10 8 (1.0)	H11 5 (0.7)	H12 5 (1.6)
			**Elevated WBC**	I1 8 (1.2)	I2 6 (1.0)	I3 1 (0.5)	I4 9 (0.8)	I5 5 (1.0)	I6 1 (0.2)	I7 9 (0.5)	I8 9 (0.3)	I9 8 (1.8)	I10 9 (0.5)	I11 5 (1.1)	I12 6 (1.9)
		**Afebrile**	**Neutropenia**	J1 2 (2.0)	J2 2 (1.5)	J3 1 (0.0)	J4 3 (2.1)	J5 3 (1.4)	J6 1 (0.1)	J7 5 (2.2)	J8 5 (2.2)	J9 5 (2.8)	J10 7 (2.1)	J11 4 (1.5)	J12 3 (2.2)
			**Normal WBC**	K1 1 (0.4)	K2 1 (0.3)	K3 1 (0.0)	K4 1 (0.3)	K5 1 (0.3)	K6 1 (0.0)	K7 1 (1.5)	K8 1 (1.2)	K9 5 (2.9)	K10 1 (2.0)	K11 1 (1.0)	K12 3 (2.2)
			**Elevated WBC**	L1 2 (1.6)	L2 5 (0.8)	L3 1 (0.0)	L4 3 (1.7)	L5 5 (0.5)	L6 1 (0.0)	L7 5 (1.7)	L8 5 (1.2)	L9 5 (2.5)	L10 5 (1.8)	L11 5 (1.2)	L12 5 (2.0)
**Elevated bilirubin**	**Normal ALT/AST**	**Febrile**	**Neutropenia**	M1 9 (0.1)	M2 2 (1.4)	M3 8 (1.2)	M4 9 (0.0)	M5 2 (1.5)	M6 8 (0.9)	M7 9 (0.0)	M8 1 (1.2)	M9 9 (0.0)	M10 9 (0.0)	M11 1 (1.5)	M12 9 (0.2)
			**Normal WBC**	N1 7 (1.1)	N2 5 (0.9)	N3 8 (1.0)	N4 8 (1.0)	N5 5 (1.1)	N6 8 (1.1)	N7 9 (0.9)	N8 4 (1.2)	N9 9 (0.0)	N10 9 (0.5)	N11 3 (1.6)	N12 9 (0.2)
			**Elevated WBC**	O1 9 (0.5)	O2 5 (1.6)	O3 9 (0.5)	O4 9 (0.9)	O5 5 (1.6)	O6 9 (0.6)	O7 9 (0.5)	O8 4 (2.1)	O9 9 (0.6)	O10 9 (0.5)	O11 1 (1.5)	O12 9 (0.2)
		**Afebrile**	**Neutropenia**	P1 5 (1.5)	P2 5 (0.9)	P3 7 (1.2)	P4 5 (1.6)	P5 4 (1.3)	P6 6 (1.4)	P7 7 (2.1)	P8 5 (1.8)	P9 9 (0.1)	P10 6 (2.0)	P11 4 (1.2)	P12 9 (0.8)
			**Normal WBC**	Q1 2 (1.5)	Q2 2 (1.8)	Q3 6 (1.2)	Q4 2 (1.8)	Q5 2 (1.5)	Q6 5 (1.2)	Q7 3 (2.2)	Q8 5 (2.1)	Q9 9 (0.3)	Q10 3 (2.0)	Q11 3 (1.8)	Q12 9 (1.0)
			**Elevated WBC**	R1 5 (1.4)	R2 5 (1.1)	R3 7 (1.0)	R4 5 (1.5)	R5 5 (1.5)	R6 7 (1.4)	R7 7 (1.8)	R8 5 (1.7)	R9 9 (0.2)	R10 7 (1.7)	R11 5 (1.6)	R12 9 (0.9)
	**Elevated ALT/AST**	**Febrile**	**Neutropenia**	S1 9 (0.1)	S2 2 (1.7)	S3 8 (0.8)	S4 9 (0.0)	S5 2 (1.7)	S6 8 (1.0)	S7 9 (0.0)	S8 1 (1.2)	S9 9 (0.0)	S10 9 (0.0)	S11 1 (1.5)	S12 9 (0.3)
			**Normal WBC**	T1 8 (0.8)	T2 5 (1.1)	T3 8 (1.4)	T4 9 (0.8)	T5 5 (1.3)	T6 8 (0.9)	T7 8 (0.9)	T8 4 (1.1)	T9 9 (0.0)	T10 9 (0.5)	T11 4 (1.7)	T12 9 (0.3)
			**Elevated WBC**	U1 9 (0.5)	U2 5 (1.5)	U3 8 (0.5)	U4 9 (0.6)	U5 4 (1.8)	U6 9 (0.7)	U7 9 (0.5)	U8 3 (1.8)	U9 9 (0.0)	U10 9 (0.5)	U11 1 (1.5)	U12 9 (0.2)
		**Afebrile**	**Neutropenia**	V1 5 (1.4)	V2 5 (1.3)	V3 7 (1.2)	V4 5 (2.0)	V5 4 (1.3)	V6 7 (1.5)	V7 7 (2.0)	V8 5 (1.8)	V9 9 (0.0)	V10 7 (2.2)	V11 3 (1.5)	V12 9 (0.6)
			**Normal WBC**	W1 3 (1.6)	W2 3 (1.7)	W3 7 (1.5)	W4 3 (2.4)	W5 3 (1.9)	W6 6 (1.5)	W7 3 (2.5)	W8 3 (1.6)	W9 9 (0.2)	W10 3 (2.2)	W11 3 (2.0)	W12 9 (0.8)
			**Elevated WBC**	X1 5 (1.6)	X2 5 (1.2)	X3 7 (1.2)	X4 7 (1.8)	X5 5 (1.8)	X6 7 (1.3)	X7 7 (1.8)	X8 5 (1.5)	X9 9 (0.1)	X10 7 (1.7)	X11 4 (1.7)	X12 9 (0.8)

In a patient being actively-treated with chemotherapy (i.e., ≤3 weeks from last chemotherapy) or with liver-directed therapy, internal/external radiation, or endoscopic stent procedure, or who received chemotherapy >3 weeks prior to presentation. Light Grey: Appropriate, Grey: Uncertain, Dark Grey: Inappropriate, Black: Disagreement Value in bold, Median: Mean deviation from median.

**Table 3 cancers-12-02375-t003:** Number of scenarios scored as ‘inappropriate’, ‘uncertain’, ‘appropriate’, or ‘disagreement’. *

Frequency of Agreement in Table 2			
	N (%)	Cumulative Frequency	Cumulative Percent
Inappropriate	87 (30.2)	87	30.2
Uncertain	73 (25.3)	160	55.5
Appropriate	98 (34.0)	258	89.5
Disagreement	30 (10.4)	288	99.9

* Definition of Scoring Agreement: Indications were classified into three levels of panel agreement or a single level for disagreement, using the following definitions: ***Appropriate***: panel median of 7–9, without disagreement, ***Uncertain***: panel median of 4–6, without disagreement, ***Inappropriate***: panel median of 1–3, without disagreement, ***Disagreement***: >2 responses in 1–3 range AND >2 responses in 7–9 range. If there were an even number of panelists, decimal medians were indicated (e.g., 3.5, 6.5) and rounded up to roll into an appropriateness level.

**Table 4 cancers-12-02375-t004:** Average panel median and average mean absolute deviation from median.

Median Ratings in Table 4					
	N	Mean	SD	Min	Max
Average of Median Ratings	288	5.22	2.81	1.00	9.00
Average of Mean Deviations from the Medians	288	1.10	0.72	0.00	2.92

**Table 5 cancers-12-02375-t005:** Guidelines.

	In pts with Elevated Bilirubin		
	**Appropriate**	**May be appropriate**	**Inappropriate**
Stent/Tube manipulation	Yes		
Inpatient Antibiotics	If pt is febrile	If the pt is afebrile but has an elevated WBC or is neutropenic	If the pt is afebrile and has a normal WBC
	In pts with normal bilirubin		
	**Appropriate**	**May be appropriate**	**Inappropriate**
Stent/Tube manipulation	Yes, if the pt has new or worsening biliary dilatation		
Inpatient Antibiotics	If pt is febrile	If the pt is afebrile but has a new or worsening biliary dilatation and has an elevated WBC or is neutropenic	If the pt is afebrile and has a normal WBC

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
