# Peer review of "Guidelines for Management of Urgent Symptoms in Patients with Cholangiocarcinoma and Biliary Stents or Catheters Using the Modified RAND/UCLA Delphi Process"

_cancers, 2020, doi:10.3390/cancers12092375_

Round 1

Reviewer 1 Report

This guideline for management of urgent symptoms of acute cholangitis in patients with cholangiocarcinoma was prepared by Delphi consensus meeting of many experts in 2016.

The aim of this study to guide physicians to appropriate decision of intervention in patients with cholangiocarcinoma who are happened to with urgent symptoms suspicious of cholangitis. It might be a very unique and clinically implicative study.

1, To introduce these conclusions, there have been not so many evidences whether these recommended intervention procedures are really appropriate or not on the basis of patient`s outcome if these interventions are utilized in patients with cholangiocarcinoma.

It might be better to mention about that clinical benefits obtained by applying recommended interventions according to this guideline is still uncertain in “discussion session”.

2, In NO 3 of “Consensus statements” it was shown that “ Antibiotics may be appropriate if the patient is afebrile but has new or worsening biliary dilatation and is neutropenic or has elevated WBC. In this content, judgement of “new or worsening biliary dilatation” is a little bit uncertain. I think that it is much better how this finding should be suspected. What kinds of imaging tool should be utilized and which findings should be carefully identified? These more practical information should be added to this guideline for more useful guideline.

Reviewer 2 Report

This study addresses the management of urgent symptoms in patients with cholangiocarcinoma and biliary stents. I have the following remarks:

1) The Authors state that  "the minimum required data for decision making when a patient presents with urgent symtoms with biliary stents was the availability of stent manipulation, basic blood tests and imaging to assess if the patient has new or worse biliary duct dilatation..". In addition it is relevant the ECOG status. By contrast, the tumor staging is unnecessary, and I wonder why it was not included in the theoretical framework, considering that imaging techniques and comparison with previous data are part of the decision process. In clinical scenarios Q1-Q6 and W1-W6 it would be particularly helpful to evaluate the tumor progression. (Chaiteerakij et al 2014).

Candida albicans and Candida parapsilosis infections are frequently reported along with microbial infections of the biliary stent. Is antifungal treatment appropriate in febrile patients? Are biliary cultures mandatory in targeted antibiotic therapy?

Round 2

Reviewer 1 Report

Nothing to be required for further revision to authors.